# Selective self-assembly of 2,3-diaminophenazine molecules on MoSe$_2$ mirror twin boundaries

Xiaoyue He [1], Lei Zhang[1], Rebekah Chua[1,2], Ping Kwan Johnny Wong[3], Arramel Arramel[1], Yuan Ping Feng [1], Shi Jie Wang[4], Dongzhi Chi[4], Ming Yang [4], Yu Li Huang [1,4] & Andrew Thye Shen Wee [1,3]

The control of the density and type of line defects on two-dimensional (2D) materials enable the development of new methods to tailor their physical and chemical properties. In particular, mirror twin boundaries (MTBs) on transition metal dichacogenides have attracted much interest due to their metallic state with charge density wave transition and spin-charge separation property. In this work, we demonstrate the self-assembly of 2,3-diaminophenazine (DAP) molecule porous structure with alternate L-type and T-type aggregated configurations on the MoSe$_2$ hexagonal wagon-wheel pattern surface. This site-specific molecular self-assembly is attributed to the more chemically reactive metallic MTBs compared to the pristine semiconducting MoSe$_2$ domains. First-principles calculations reveal that the active MTBs couple with amino groups in the DAP molecules facilitating the DAP assembly. Our results demonstrate the site-dependent electronic and chemical properties of MoSe$_2$ monolayers, which can be exploited as a natural template to create ordered nanostructures.

[1] Department of Physics, National University of Singapore, 2 Science Drive 3, Singapore 117542, Singapore. [2] NUS Graduate School for Integrative Sciences & Engineering (NGS), National University of Singapore, 28 Medical Drive, Singapore 117456, Singapore. [3] Centre for Advanced 2D Materials (CA2DM) and Graphene Research Centre (GRC), National University of Singapore, Singapore 117546, Singapore. [4] Institute of Materials Research & Engineering (IMRE), A*STAR (Agency for Science, Technology and Research), 2 Fusionopolis Way, Innovis, Singapore 138634, Singapore. Correspondence and requests for materials should be addressed to M.Y. (email: yangm@imre.a-star.edu.sg) or to Y.L.H. (email: chocosea@gmail.com) or to A.T.S.W. (email: phyweets@nus.edu.sg)

Two-dimensional (2D) transition metal dichalcogenides (TMDs) have great potential for future device applications due to their novel electronic and optical properties[1]. These properties are often strongly modified by defects[2–6], which are inevitable in the growth process. As one of the most common defects, grain boundaries (GBs), have been widely observed in monolayer TMDs synthesised by bottom-up approaches such as chemical vapour deposition (CVD) and molecular beam epitaxy (MBE)[7–10]. Bandgap tunability has been observed in single-layer $MoS_2$ GBs[11], and mid-gap states reported at low-angle GBs in single-layer $WSe_2$[12]. Other irregular or regular line defects with metallic properties containing 4-, 5-, and 8-membered rings have also been reported in $MoS_2$, $WS_2$, and $WSe_2$ in contrast to the semiconducting nature of their pristine 2D crystals[13]. More recently, mirror twin boundaries (MTBs) in atomically thin $MoSe_2$ films have attracted increasing attention due to the direct observation of one-dimensional (1D) charge density waves (CDW) along the MTBs at low temperatures[14]. Dense MTB networks are observed in MBE-grown $MoSe_2$ or $MoTe_2$ mono-layers on graphene (and also graphite) or $MoS_2$ surfaces, and their atomic structures and electronic properties have been investigated by scanning probe microscopy and photoemission spectroscopy[15,16]. Such metallic line defects are usually unfa-vourable for carrier/energy transport and optical applications. However, they are often catalytically active due to their large density of states (DOS)[17–20], and may potentially be used as templates for self-assembly of molecules[21,22], as site-selective adsorption behaviours have been previously reported in 2D materials with a variety of surface inhomogeneities, including edge sites[18], atomic defects[23], moiré patterns[24,25], intrinsical 1 H/1 T patterns[26], and so on.

This work elucidates the atomic structures and electronic properties of MTBs in single-layer (SL) $MoSe_2$ grown on graphite, and investigates their chemical reactivity with organic molecules. Using a combination of ultrahigh vacuum scanning tunnelling microscopy/spectroscopy (STM/STS) and non-contact atomic force microscopy (nc-AFM) techniques, we image the electronic modulations in the (quasi) periodic MTB structures, revealing their metallic nature and lower surface potential (work function). 2,3-diaminophenazine (DAP), an amino derivative of phenazine with promising luminescence, electrochemical and biochemical applications[27–30], is selected for investigating the template effect of the dense MTB network. We observed the formation of a porous structure of DAP molecules that map onto the wagon-wheel patterns of the underlying $MoSe_2$. First-principles calcu-lations further suggest that the active amino groups in the DAP molecules play a critical role in this preferential adsorption behaviour, as charge redistribution occurs mainly close to the amino groups. By comparing the adsorption energies of different adsorption configurations, the configuration with the DAP molecule adsorbed parallel to and on the top of the MTBs is found to be the most stable, in agreement with the experimental results. This study demonstrates that organic molecule self-assembly can be facilitated by domain boundaries in epitaxial 2D TMDs. These defective TMDs and porous organic molecule structures have potential applications in site-selective catalysis, or as molecular sensors or flexible organic optoelectronic devices.

## Results

### Atomic structure and electronic properties of MBE-grown $MoSe_2$ films on HOPG.
Figure 1a shows a typical large-scale STM image of a $MoSe_2$ film grown by MBE on a highly oriented pyrolytic graphite (HOPG) substrate. The whole $MoSe_2$ surface is decorated by triangle patterns. A close-up STM image of these patterns is displayed in Fig. 1c, and the boundary appears as two bright parallel rows at negative sample voltages (occupied states). Hence the single layer $MoSe_2$ film is comprised of a mosaic of triangle single-crystalline domains, where neighbouring domains are separated by inversion domain boundaries (IDBs) or mirror twin boundaries (MTBs). These triangle patterns are non-uniform in size, with sizes varying from several to tens of nanometres (nm). Fig. 1b shows the statistical distribution of the triangle domain size (represented by the side lengths of the MTB equilateral triangles) of more than 300 triangles in the samples studied. It is found that these $MoSe_2$ triangle domains are mostly in the size range of 1.5–4.0 nm, which corresponds to 5–12 lattice constants of $MoSe_2$ ($a_0 = 3.3$ Å as determined by nc-AFM mea-surements). The smaller triangle domains tend to pack in the centre of the $MoSe_2$ islands while the larger domains are observed closer to the island edges.

The detailed atomic structures of the MTBs are further studied by atomically resolved nc-AFM, which eliminates interference from electron density of states imaged in the STM topological images. The nc-AFM measurements were carried at 77 K with a native tungsten tip without CO functionalization. Figure 1d shows the nc-AFM image of the same area as the STM image in Fig. 1c with high resolution. In this large-scale AFM image ($30 \times 30$ nm²), the triangle $MoSe_2$ domains are uniform in contrast, and the MTBs are easily distinguishable as line defects with darker contrast. In frequency-shift nc-AFM images which are sensitive to the tip-sample separation as well as the local electron densities, positive $\Delta f$ is generally a result of short-range Pauli repulsive interactions, while negative $\Delta f$ is due to long-range attractive van der Waals interactions and/or electrostatic forces[31]. Therefore, the observed darker contrast with negative frequency shift in the MTB regions might be due to their relatively lower height and/or rich electron densities in comparison to the $MoSe_2$ triangle domains.

The precise atomic structures are clearly revealed in the high-resolution nc-AFM image of Fig. 1e. The corresponding atomistic model is displayed in Fig. 1f. Based on the nc-AFM working principles and by comparing with reported atomic structures of $MoSe_2$ triangle domains[14,32], the hexagonal lattices with bright contrast (orange) are attributed to the top-layer Se atoms, which are closer to the tip and generate repulsive forces. The dark features (blue) are attributed to the lower-lying Mo atoms, which are further from the tip and experience attractive forces. The MTBs consist of a Se atomic row sandwiched by two Mo atomic rows connecting two adjacent inverted $MoSe_2$ triangle domains with Se-edges (red dashed line). The darker contrast in the MTB regions is due to higher density of Mo atoms, where each Se atom is bound to four Mo atoms instead of three, and thus giving rise to stronger attractive forces. The darkest contrast around the intersection of the wagon-wheel patterns is attributed to Mo vacancies. The separation between Se-edges of two neighbouring inversion domains is ~0.59 nm, close to $\sqrt{3}a_{MoSe2}$, as marked by a white arrow in Fig. 1e. This is consistent with previously reported values from theoretical calculations, transmission electron microscopy results[33,34], and nc-AFM studies with CO-functionalized tips[14]. The atomic structures confirmed by the STM/nc-AFM measurements facilitate a better understanding of the modulation of the electronic structures in the MTBs, which will be discussed below.

Figure 2a shows a highly-resolved STM topological image of the SL-$MoSe_2$ surface, where the MTBs appear as bright protrusions with periodic intensity undulations due to charge density wave (CDW) modulation at low temperature[14]. From the lateral profile shown in Fig. 2b, the periodicity is determined to be $1.00 \pm 0.03$ nm, equivalent to ~3 lattice constants of $MoSe_2$, consistent with that reported in the literature[14,16,32,34]. Typical STS spectra taken on the pristine $MoSe_2$ triangle domain and the MTBs are shown in Fig. 2c. The red curve clearly reveals

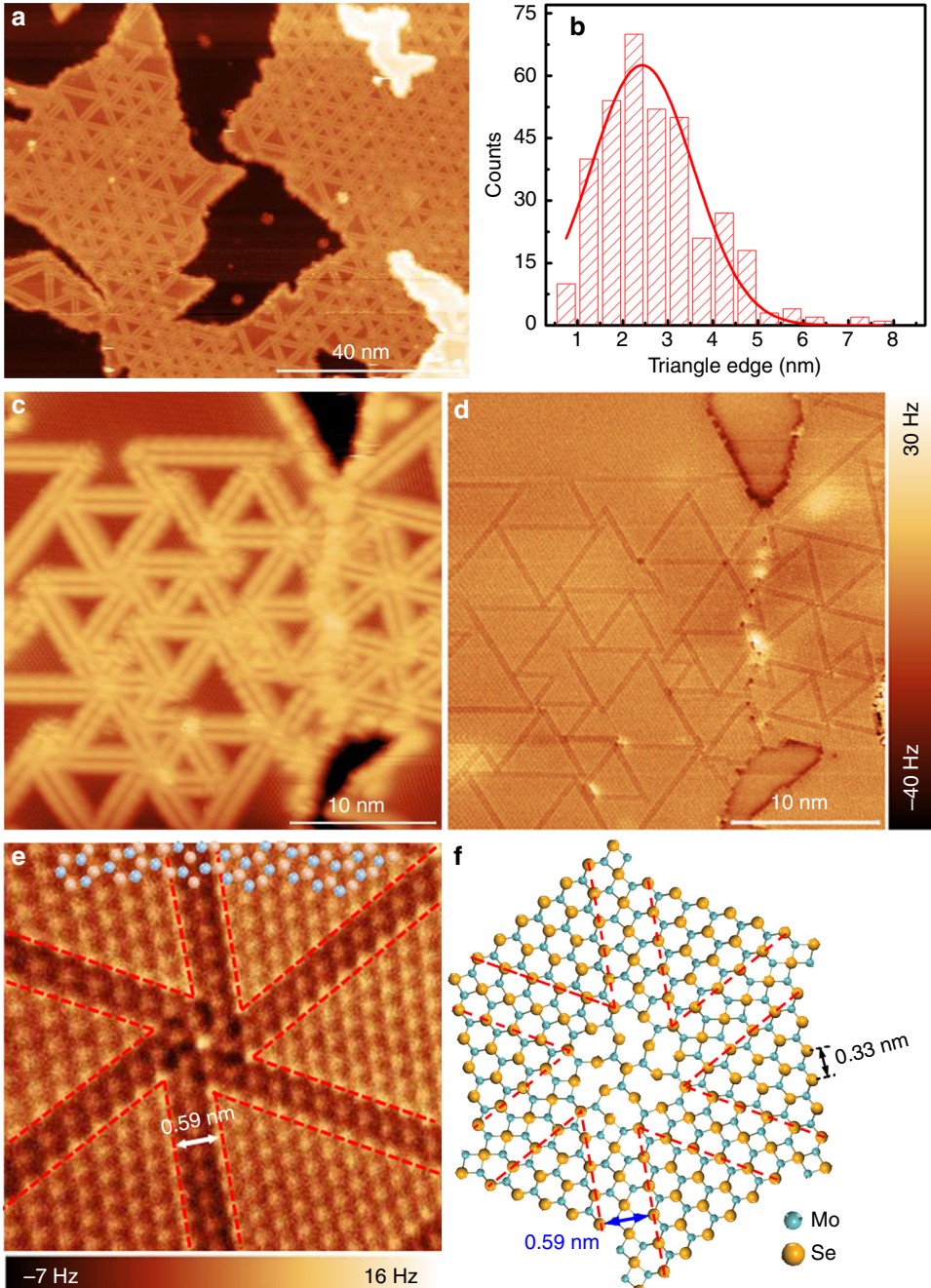

**Fig. 1** The topographic structures of MBE-grown MoSe₂ films. **a** Large-scale STM image of 0.85 ML MoSe₂ on HOPG (size: $100 \times 80$ nm², $V_S = -1.4$ V). **b** Statistical analysis of the triangle MoSe₂ domain size. The red line is the fitting distribution curve. **c** A close-up STM image of the inversion domain boundary network (size: $30 \times 30$ nm², $V_S = -1.3$ V). **d** The corresponding nc-AFM image recorded in the same area as panel **c**. **e** A high-resolution nc-AFM image (size: $5 \times 5$ nm²) shows a typical wagon-wheel pattern. The Se atoms appear as bright spots in the frequency shift image. Here, the orange and blue balls represent the Se and Mo atom, respectively. **f** An atomistic model corresponding to the experimental structure shown in **e**. The red dashed line highlights the Se-edge of MoSe₂ triangle domains

semiconductor characteristics of the single-layer MoSe₂ away from the MTBs, with an electronic bandgap of $2.0 \pm 0.1$ eV[15]. In contrast, the blue d$I$/d$V$ spectrum acquired on the MTB shows distinct electronic states inside the bandgap. In particular, a "V" shaped feature around the Fermi level (0 V) (highlighted by a rectangle in Fig. 2c) is observed, indicating the metallic nature of the MTBs[5,9,15,32,35]. Thus, the mosaic MoSe₂ surface is composed of alternating semiconducting and metallic regions, which might enable unique site-selective behaviour at the nanoscale[18,26], e.g., for adsorption, catalysis, and so on.

Another key parameter that determines the surface electronic properties is the local work function or surface potential, which can be measured with high spatial resolution by field emission resonances (FERs) at the STM tip-sample junction[36,37]. FERs are also known as Gundlach oscillations[38] in the Fowler-Nordheim tunnelling regime. When the applied bias is large enough to lift the tip Fermi level above the sample vacuum level, hydrogen-like electronic resonances occur in the vacuum junction between the tip and the sample[39–41]. As thus, the first resonance peak of the FERs measured by STS (e.g. d$I$/d$V$ and d$z$/d$V$) is related to the

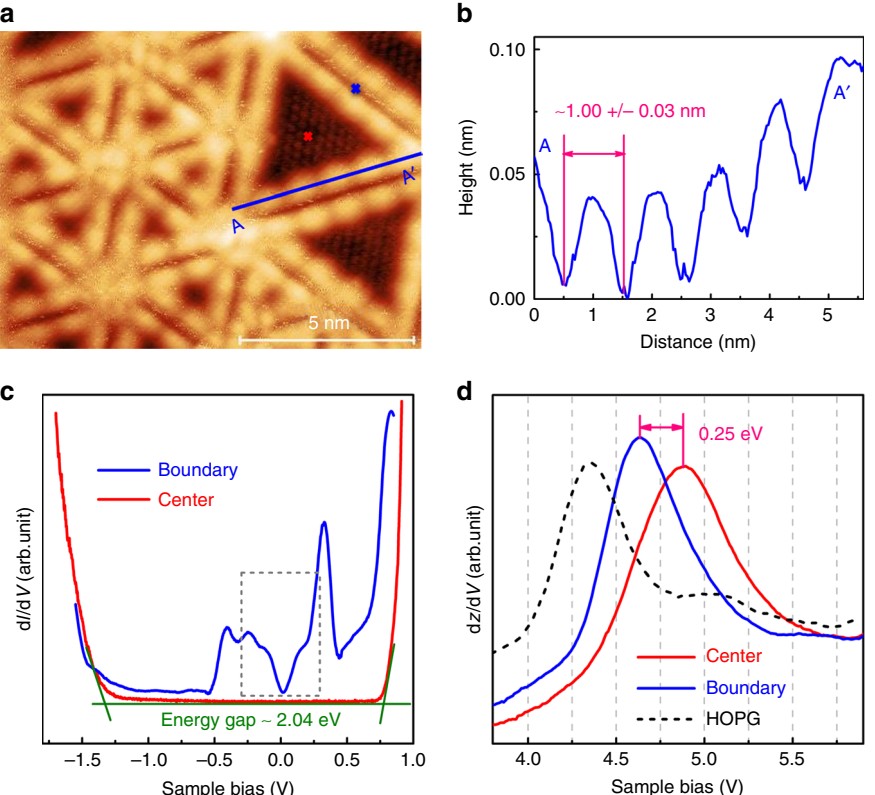

**Fig. 2** The electronic properties of the MoSe$_2$ film. **a** Atomically-resolved STM image of the ML-MoSe$_2$ surface shows the periodic intensity undulations in MTBs ($V_S$ = -900 mV and $I$ = 96 pA; size: 12 × 9 nm$^2$). **b** The line profile along the blue line AA' in figure (**a**). **c** d$I$/d$V$ spectra recorded at the domain centre (red line) and MTB boundary (blue line) reveal their semiconducting and metallic nature. **d** Field emission resonances (FERs) measured by d$z$/d$V$ spectra on HOPG substrate, MTBs and the centre of triangle MoSe$_2$ domains, respectively. Data were taken at constant tip height ($V_S$ = 1.2 V, $I$ = 60 pA) with a modulation of 25 mV and 963 Hz. The red and blue crosses in panel **a** mark the locations of the measurements of the spectra shown in **c** and **d**

local work function of the underlying sample. Fig. 2d shows the FER spectra measured at different regions of MoSe$_2$ as well as the clean HOPG substrate for comparison. On the bare HOPG (black dashed curve), the first FER peak appears at 4.35 ± 0.05 eV, in a good agreement with the work function of HOPG in the range of 4.4–4.8 eV[42]. Interestingly, the triangle MoSe$_2$ domains (red curve) yield their first FER at 4.88 ± 0.05 eV, while the MTB regions (blue curve) first resonate at 4.63 ± 0.05 eV, indicating a shift of ~0.25 eV towards lower energy compared to that of the adjacent pristine triangle domain.

**Formation of DAP porous structure on MoSe$_2$ surface**. To explore the possible site-specific properties, we deposited an amino derivative of phenazine, 2,3-diaminophenazine (DAP), onto the MoSe$_2$ surface. The molecular structure of DAP is shown in the inset of Figure 3a (Supplementary Fig. 1 in supplementary information).The structure of the DAP molecule is essentially planar with two active amino-groups at one end. Upon deposition of a sub-monolayer of DAP molecules, the supramolecular structure networks formed are obvious and distinguishable, but their sizes and shapes are not uniform as shown in Fig. 3a, b. Fig. 3a suggests that the DAP molecules mainly adsorb on the edges of the second and the centre of the first layer MoSe$_2$ at lower coverage. The adsorbed DAP area expands to fill the whole ML-MoSe$_2$ surface with increasing coverage as shown in Fig. 3b. The porous structure sizes can reach hundred nanometres (Supplementary Fig. 2) and are only limited by the MoSe$_2$ island sizes. Figure 3c displays a typical DAP pore region. The shapes of the pores are slightly distorted from perfect hexagons. Statistical

analysis of the pore sizes in Fig. 3d reveals the distribution of pore size is mainly in the range of 1.5–4.0 nm. This coincides with the size distribution of the MoSe$_2$ triangle domains (the red dashed line), although the peak has a small offset due to the offset of MTBs from the centre of the wagon-wheel patterns (most of the two opposite MTBs move several atomic lattices relative to each other, seen Fig. 1e, f). These results suggest that the DAP assembled structures map directly onto the wagon-wheel patterns of the underlying MoSe$_2$ substrate.

Figure 4a shows a high-resolution STM image of the DAP nanoporous structure with a partially uncovered MoSe$_2$ surface. Each DAP molecule appears as a dumbbell-shape feature with slightly asymmetric contrast along the long axis direction. This is understandable due to its asymmetric molecular structure (amino-groups at one end). By a statistical analysis of the orientation distribution of the DAP molecules with respective to the MTBs (Supplementary Fig. 4), it is found that the DAP molecules oriented with their long axes parallel to the MTBs are dominant. The alignment of the molecular orientations with the three high symmetry directions of MoSe$_2$ indicating a significant substrate effect[43].

In the porous structure, two typical packing configurations are observable, namely linear configurations (L-type) and triangle trimer configurations in a head-to-tail arrangement (T-type) highlighted by the green circles and red triangles in Fig. 4a, b respectively. We have analysed ~1000 DAP molecules to understand the distributions of the L-type and T-type configurations (the statistical histogram is shown in Supplementary Fig. 5). The percentage of L-type configurations is slightly higher than that of the T-type. The L-type configuration preferentially forms on the

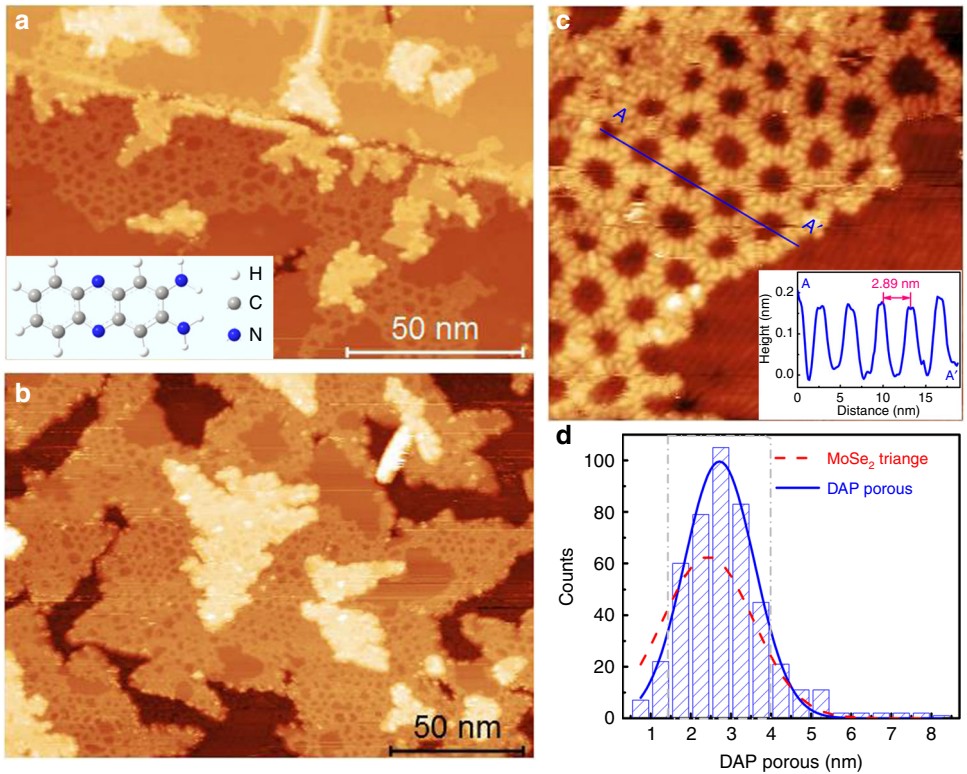

**Fig. 3** STM images of the sub-monolayer DAP on MoSe₂/HOPG. **a** 0.36 ML and **b** 0.80 ML DAP molecules assemble on the MoSe₂ surface, forming porous supramolecular structure. (size: **a**, 180 × 120 nm²; **b**, 200 × 150 nm²; $V_S$ = 2.2 V, $I$ = 86 pA). The inset in panel **a** is a DAP molecular structure. **c** An enlarged STM image of the DAP porous network (size: 30 × 30 nm²; $V_S$ = 2.75 V, $I$ = 100 pA).The inset is the line-profile corresponding to the blue line in panel **c**. **d** Statistical analysis of the DAP porous size over many large-scale images (more than 400 DAP pores). The blue line and the red dash line are the fitting distribution curves of DAP porous and triangle MoSe₂ domain size, respectively. The grey dash-dot-line shows the main distribution region of the size

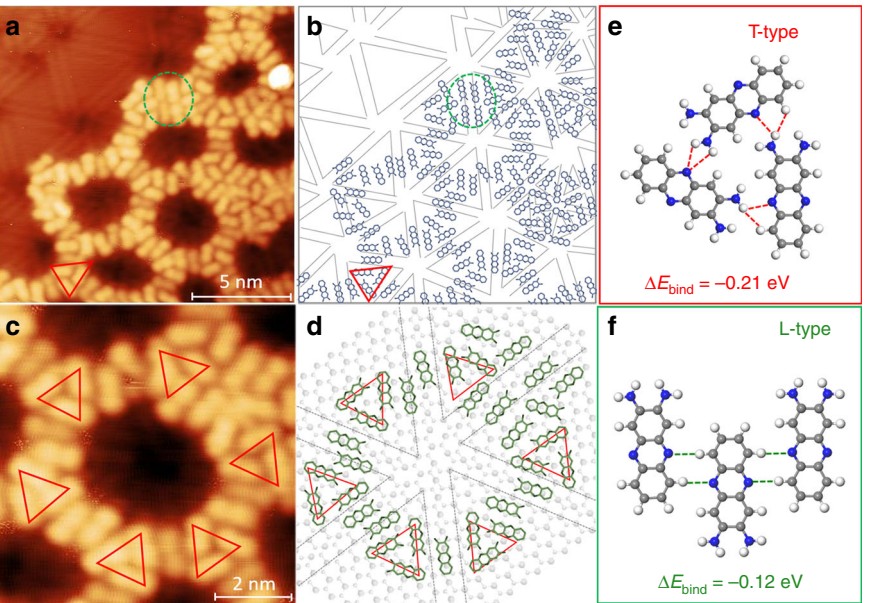

**Fig. 4** Model of DAP porous configurations on MoSe₂ surface. **a** High-resolution STM image (15 × 15 nm²; $V_S$ = 1.7 V, $I$ = 60 pA) of DAP porous structure on MoSe₂. **b** Schematic of the DAP molecular configuration described in **a**. The red triangle shows a typical triangle trimer configuration (T-type) in **e**, and the green circle shows the linear configuration (L-type) of DAP molecules in **f**. **c** A close-up STM image (Size: 8 × 8 nm²) of a prototypical DAP pore. **d** The atomic configuration of a typical DAP hexagonal pore on the top of a wagon-wheel pattern. The red triangle highlights the T-type trimer configuration, which act as the corner of the hexagonal pores. Here, the binding energy per DAP molecule in trime (**e**) or linear (**f**) configuration is defined by
$$\Delta E_{bind} = \left( E_{total} - 3E_{single\,DAP} \right)/3$$

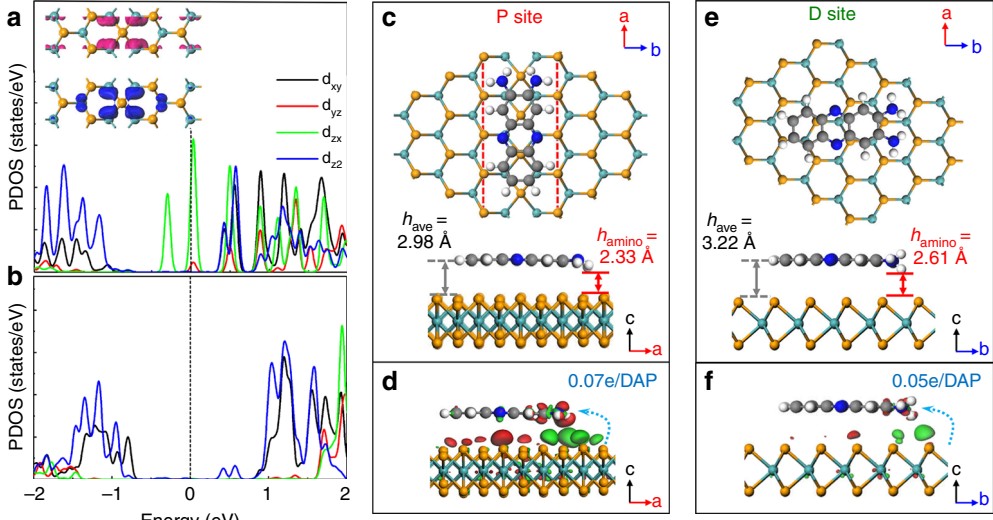

**Fig. 5** Optimised configurations of DAP molecules adsorbed on single-layer MoSe$_2$ surface. **a, b** The projected density of states of Mo atoms in the MTB and far away from the MTB (only spin-up states are shown here). The inset in **a** is the partial charge density of electronic states near Fermi level (pink and blue colours denote the states in the range of [0–0.3] eV and [−0.3-0] eV respectively) with an isosurface value of $2.0 \times 1.0^{-4}$ e/Å$^3$. **c, e** Top view and side view of DAP adsorbed on MTBs with a parallel orientation to the MTB (P site), and defect-free MoSe$_2$ basal plane (D site). $h_{ave}$ and $h_{amino}$ are the equilibrium distance from the centre of DAP molecule or amino group to the top-layer Se of MoSe$_2$ surface. **d, f** Charge density difference of P site and D site calculated by $\rho = \rho_{MoSe2+DAP} - \rho_{MoSe2} - \rho_{DAP}$. The red and green distributions denote the electron accumulation and depletion regions, respectively. The isosurface value of the visualised charge density was set to be $4.0 \times 1.0^{-4}$ e/Å$^3$. The blue dash-line shows the charge transfer from MoSe$_2$ to DAP molecule

MTB regions, while the T-type configuration preferentially forms on the MoSe$_2$ triangle domains. A typical hexagonal DAP porous nanostructure shown in Fig. 4c mimics the features of the MoSe$_2$ wagon-wheel pattern, and the corresponding configuration model is shown in Fig. 4d. The hexagonal DAP porous structure form surrounding the intersection of wagon-wheel patterns by alternatively absorbing the L-type and T-type configurations. Due to the irregularities in the hexagonal MTB network and the size mismatch between the DAP T-configurations (2.1 ± 0.1 nm) and the MoSe$_2$ triangle domains (1.5–4.0 nm), the size and shape of DAP pores are also non-uniform, and display a small distribution offset from the MoSe$_2$ MTB networks as shown in Fig. 3d.

The porous DAP molecular structure is distinctly formed by a site-specific self-assembly: preferential T-type adsorption on the defect-free MoSe$_2$ domains and L-type on the MTBs, while leaving the MTB intersections empty. With increasing molecular coverage, a close-packed phase will emerge coexisting with the loose-packed porous phase (Supplementary Fig. 6). The evolution of the DAP packing structure with annealing temperature have also been studied. We find that the DAP porous nanostructures do not become more ordered after thermal annealing, but desorb above 90 °C (Supplementary Fig. 7). The close-packed configuration starts to desorb at 140 °C. This porous nanostructure is not observable for the DAP molecules grown on graphite and Au (111) substrates (Supplementary Figs. 8 and 9). Therefore, the formation of the hexagonal DAP porous structures is steered by the underlying MTBs and the relatively weak selective adsorption, and is attributed to the subtle competition between the relatively weak DAP–substrate and DAP–DAP interactions.

**The formation mechanism of the DAP porous structure.** To gain further insight into the formation mechanism of the DAP porous structure, theoretical calculations based on density functional theory (DFT) were performed to investigate the templating effect of the MTB network. It is noted that the electronic structure of MoSe$_2$ can be tuned by external influences such as strain,

defects, or substrates[35,44–47]. Here, the presence of MTBs in the MoSe$_2$ single-layer reduces the bonding coordination of Mo atoms from 6 to 4, resulting in modified electronic and chemical properties. First-principles calculations show that the electronic structure in the MTB region (Fig. 5a) is significantly different from the defect-free region far away from the MTB (Fig. 5b), consistent with the STS measurements in Fig. 2. The unoccupied electronic states of Mo d orbitals (mainly $d_{zx}$) in the MTB spread into the band gap, forming metallic states, as shown in Fig. 5a. These slightly delocalised Mo $d_{zx}$ orbitals enhance the interaction with the adsorbed molecules, rendering the MTB more chemically active. This reactivity guides the DAP self-assembly on the MTBs as observed in the STM experiments.

Indeed, compared with DAP on pristine MoSe$_2$ single-layer, the interaction between DAP and the MTB is enhanced, as evidenced by the shorter interfacial spacing, stronger charge redistribution, and larger adsorption energy. As Fig. 5c shows, the spacing ($h_{ave}$) between the DAP benzene ring and MTB with a parallel orientation (P site) is ~2.98 Å, beyond the chemical bonding range. In contrast in Fig. 5e, a larger spacing of 3.22 Å is obtained for DAP on defect-free MoSe$_2$ (D site), indicating a physical adsorption stabilised by van der Waals (vdW) forces. From the side-views of the optimised configurations, it is worth noting that the two amine groups of the DAP molecule absorbed atop the MTB region both bend downwards ($h_{amino}$ ~ 2.33 Å), while those on the defect-free region have one amino group pointing upwards and the other pointing downwards ($h_{amino}$ ~ 2.61 Å) (Supplementary Fig. 11). Furthermore, the visualised charge density difference in Fig. 5d–f show different charge redistributions in the DAP molecule and MoSe$_2$. The charge redistribution is more pronounced at the DAP amino groups, resulting in a slightly tilted DAP molecule with a smaller spacing between amino groups and MoSe$_2$. Calculations also show that the charge transfer at the P site is slightly stronger (~0.07e per molecule) than at the D site (~0.05e per molecule) (Supplementary Fig. 12). As a consequence of the shorter interfacial spacing and the larger charge transfer, the adsorption energy of the DAP molecule adsorbed on the MTB (−1.38 eV) up to 0.16 eV is larger

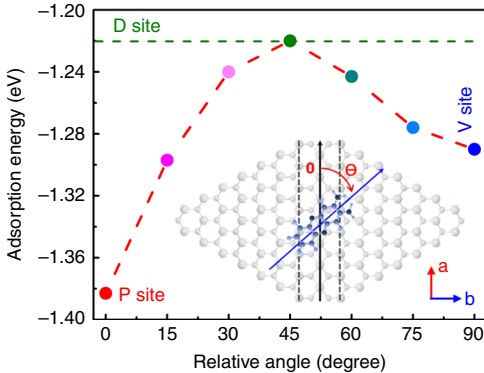

**Fig. 6** Angle-dependence adsorption energy of DAP molecule adsorbed on MTB. The inset shows the rotation angle $\theta$ of DAP relative to MTB. The green dashed line shows the adsorption energy of DAP molecule adsorbed on defect-free $MoSe_2$ basal plane (D site)

than that on the defect-free $MoSe_2$. These results suggest that the MTB on $MoSe_2$ are chemically active, facilitating the adsorption of DAP. Furthermore, the amino group is more reactive than the benzene ring, and forms the active site that electrostatically interacts with the metallic MTB.

Figure 6 shows the angle-dependence of the DAP adsorption energy on the MTB. As shown in the inset in Fig. 6, angle $\theta$ defines the orientation between the long axis of the DAP molecule and the MTB axis (a axis). It can be seen that the DAP molecule adsorbed parallel (P site) to the MTB is the most energetically favourable, with the lowest binding energy of $-1.38$ eV. This is because the electronic overlap between DAP and $MoSe_2$ is the largest in this configuration, leading to stronger adsorption. It explains why the L-type configuration is the more observed pattern on the MTBs (in Fig. 4d). Interestingly, the adsorption energy for DAP perpendicular ($\theta = 90^{\circ}$) to the MTB (or V site, as shown in Supplementary Fig. 11) has the second lowest value ($-1.29$ eV). The adsorption energy of $-1.22$ eV at $\theta = 45^{\circ}$ is equivalent to the value of the D site. This trend can be explained by the chemical reactivity of the DAP amino-groups. When $\theta$ is >45$^{\circ}$, the amino-groups detach from the MTBs and the DAP adsorption becomes stable again. The relatively small energy difference (<0.16 eV) between the most stable configuration and the others is consistent with the molecular orientation distribution observed in Fig. 4 and Supplementary Fig. 4. It should be noted that the chemical reactivity of MTBs is weaker compared with other defects in the $MoSe_2$ single-layer such as Se vacancies, due to the absence of dangling bonds. This is why the MTBs only weakly affect the orientation of DAP molecules, but the interaction is sufficient to guide DAP assembly to form hexangular nano-pores.

Finally, the formation of the DAP porous structure, particularly the coexistence of T-type and L-type building configurations in Fig. 4, can be understood from a comparison of their formation energies ($E_{bind}$). From the optimised models shown in Fig. 4e, f, we can see that both configurations are stabilised by the formation of intermolecular hydrogen bonding. That is, the C–N···H distance between the neighbouring DAP molecules is in the range of 2.5–2.9 Å (Supplementary Fig. 10), indicating weak hydrogen-bonding[48]. In the absence of a substrate, DFT calculations suggest that DAP molecules in the T-type configuration ($E_{bind} = -0.21$ eV) is more stable than that in the L-type configuration ($E_{bind} = -0.12$ eV) due to the stronger hydrogen bonding interactions, which is in contrast to the slightly higher distribution of the L-type configurations analysed in Supplementary Fig. 5. Hence, the self-assembly of DAP molecules on the MTB network is determined by a subtle balance of intermolecular

interactions in competition with the substrate templating effect. When the DAP molecules absorb on the defect-free regions, the T-type configuration is preferred because of the stronger intermolecular interactions. On the other hand, when the molecules diffuse to the MTBs, they bind parallel to the chemically active boundaries.

In conclusion, we demonstrate the self-assembly of a DAP molecular nano-porous structure on the dense MTB network surface of an MBE-grown $MoSe_2$ film. DAP molecules form a hexagonal pores structure that maintains the same symmetry as the $MoSe_2$ substrate as a result of the electronic and chemical inhomogeneity of the substrate surface. The DAP molecules aggregate into linear configurations at the edge of the hexagonal pores along the MTBs, and trimer triangle configurations form at the corner of the hexagonal pores on the defect-free $MoSe_2$ domains. DFT calculations reveal that the DAP amino groups and the metallic MTB are the main active interaction sites. Thus, the formation of the DAP porous structure is guided by the $MoSe_2$ MTB network together with intermolecular hydrogen interactions. This work offers a novel pathway to atomically precise molecular nano-patterning of 2D TMD surfaces, and provides a platform to explore the local chemical properties of $MoSe_2$.

## Methods

**$MoSe_2$ film preparation**. The $MoSe_2$ samples were prepared by molecular beam epitaxy (MBE) on commercial HOPG substrate in a home-built ultra-high vacuum (UHV) system. The growth chamber has a base pressure of $10^{-9}$ mbar. The HOPG substrates were cleaved in air and transferred into the loadlock chamber as quick as possible to minimise contamination, and then outgassed for 3 h at 550 °C before sample growth. Molecular beams of Se (99.99%, Sigma-Aldrich) and Mo (99.99%, Goodfellow) were generated from a standard Knudsen cell (MBE-Komponenten) and a commercially available e-beam cell (Focus), respectively. The source flux was calibrated by a flux monitor (quartz crystal monitor). The flux ratio between Mo and Se was at least 1:10, and the film deposition rate was 1.0 ML per hour. The sample temperature during growth was 400 °C. For subsequently ex situ STM measurements using a separate Scienta-Omicron LT-STM system, an amorphous Se capping layer was deposited on the sample at room temperature before it was taken out of the vacuum. This Se capping layer was then thermally desorbed in the Omicron preparation chamber prior to LT-STM measurements.

**DAP molecule depositon**. DAP (95%, Sigma-Aldrich) molecules were deposited from a Kundsen effusion cell in the loadlock chamber (base pressure is $10^{-8}$ mbar) of the Scienta-Omicron system. A typical deposition rate used was ~0.03 ML/min. Here, one monolayer (ML) refers to one full monolayer DAP in closed packed configuration. The deposition rate of DAP was calibrated by large-scale STM images. During the deposition, the $MoSe_2$ substrates were held at room temperature and the samples were subsequently transferred in situ to the STM chamber for analysis.

**STM/ nc-AFM imaging**. STM/nc-AFM and STS measurements were performed at 77 K in an Scienta-Omicron UHV system interfaced to a Nanonis controller equipped with and STM/qPlus sensor. Electrochemically etched tungsten tips were used, and bias voltages were applied to the sample. For STM imaging, the constant current mode was adopted and the tunnelling current was in the range of 60–200 pA. For STS measurements, the lock-in technique was used, with a modulation voltage of 30 mV and frequency of 963 Hz. For nc-AFM imaging, the constant-height mode with an oscillation amplitude of 1 nm was used to record the frequency shift ($\Delta f$) of the qPlus resonator (sensor frequecy $f_0 \approx 24$ kHz, $Q \approx 8000$). All the STM/STS data were analysed and rendered using WSxM software[49].

**DFT calculation**. The first-principles calculations were performed using density-functional theory (DFT) based Vienna ab initio simulation package (VASP 5.4.4.18), in which Perdew-Burke-Ernzerhof (PBE) exchange-correlation functionals and the projector augmented wave (PAW) potentials were used. The cut-off energy for the plane wave expansion was set to 500 eV. We have calculated pristine $MoSe_2$ monoalayer, $MoSe_2$ single-layer with MTB, DAP molecule, DAP molecule on the perfect $MoSe_2$ surface, and DAP molecule on the surface of $MoSe_2$ with MTB, where the corresponding first Brillouin zone was sampled by using Γ-centred $12 \times 12 \times 1$, $12 \times 1 \times 1$, $1 \times 1 \times 1$, $3 \times 3 \times 1$, and $3 \times 1 \times 1$ k-point meshes, respectively. A vacuum layer >15 Å was applied normal to the surface of $MoSe_2$ single-layer. For the interfaces between DAP molecular and $MoSe_2$, van der Waals (vdW) correction was included by using Grimme's DFT-D3 method. In all calculations, the criterion for energy and force convergence was set to $1.0 \times 10^{-6}$ eV and 0.01 eV/Å, respectively. The adsorption energy for DAP molecule on $MoSe_2$ single-layer

was estimated by:

$$\Delta E_{ad} = E_{MoSe2+DAP} - E_{MoSe2} - E_{DAP}, \qquad (1)$$

where $E_{MoSe2+DAP}$ is the total energy of the hybrid structures for DAP on $MoSe_2$, and $E_{DAP}$ and $E_{MoSe2}$ are the total energies of isolated DAP molecule and $MoSe_2$, respectively. The charge redistribution of DAP molecular on $MoSe_2$ was estimated by the integration of in-plane averaged charge density difference and Bader charge analysis.

## Data availability

The data that support the findings of this study are available from the corresponding authors upon reasonable requests.

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

## Acknowledgements

A.T.S.W. acknowledges funding support from MOE Tier 2 grant R-144-000-382-112, Tier 3 grant R-143-000-625-112, and A*STAR Pharos Program Grant No. 1527300025. Y.L.H., M.Y., S.J.W., and D.C. acknowledge the A-STAR SERC grant support for the 2D growth project under the 2D pharos program (SERC 1527000012). X.H. acknowledges a research grant from the National Natural Science Foundation of China (NSFC, Grant Nos.11404073 and 11674366). P.K.J.W. is supported by the Singapore NRF Medium Sized Centre Programme R-723-000-001-281. A. acknowledges a research grant from MOE Tier 3 programmer (MOE2014-T3-1-004).

## Author contributions

X.H., Y.L.H. and A.T.S.W. proposed and designed the research project. X.H. performed the STM experiments and analysed the data. Y.L.H. and D.C. participated in discussion and analysis of the data. Y.M. helped in discussion and performed the first-principle calculations. Z.L., R.C. and P.K.J.W. provided the MBE-grown $MoSe_2$ samples. Arramel participated in the STM measurements. Y.P.F. and W.S.J. supervised the calculations. X.H., Y.L.H., Y.M. and A.T.S.W. wrote the paper, and all authors discussed and revised the final manuscript.

## Additional information

**Competing interests:** The authors declare no competing interests.

**Peer Review Information**: *Nature Communications* thanks the anonymous reviewer(s) for their contribution to the peer review of this work. Peer reviewer reports are available.

