## [Peer Review File · Nature Communications]

Reviewers' comments:

Reviewer #1 (Remarks to the Author):

In this manuscript, Wee et al show the enhanced chemical reactivity of a special type of grain boundaries in transition metal chalcogenides (TMDs), namely the mirror-twin boundaries (MTBs) existing on MoSe₂.

UHV STM clearly shows the atomic structure, electronic property and 'local' work function of MTBs in monolayer MoSe₂. The enhanced chemical reactivity of the MTBs is evidenced by the site-selective adsorption of 2,3-diaminophenazine (DAP) either parallel to or on top of the MTBs. Thus the molecular assembly could be steered and interesting patterns generated. The STM imaging and tunneling spectroscopy are of high quality, and the conclusions are adequately supported by complementary theoretical modeling; as such the study is pertinent and state of the art.

Overall, the ms reports several interesting findings, but key claims seem to be exaggerated. Notably "site-dependent electronic and chemical properties of MoSe₂" have been recognized earlier and "large-scale ordered nanostructures" remain elusive (see the ms abstract). Also on p.2 "regular porous networks" are mentioned, although no convincing data is at hand. It also must be said that line or other types of defects have been extensively employed to steer molecular assembly (using, e.g., vicinal surfaces, strain relief patterns, anisotropic crystal orientations, surface reconstructions, graphene or h-BN modulated templates etc.). Statements like 'pioneering' or 'striking' should be omitted and tangible ideas described towards improved control and how to use the realized nanosystems for applications.

Furthermore, there are some technical issues, see comments below:

- page 3 Paragraph 2: "bright contrast (higher Δf) is generally a result of short-range Pauli repulsive interactions, while darker contrast (lower Δf) is due to long-range"; this description is not accurate.

- the AFM repulsive regime should correspond to positive Δf , and the attractive regime should correspond to negative Δf , just calling it higher or lower without mentioning the sign of Δf is misleading.

- on page 6 Paragraph 1, the authors mentioned "the adsorption energy of the DAP molecule adsorbed on the MTB is 0.16eV larger than that on the defect-free MoSe₂". Can the authors comment on how useful this energy difference is, (for example) by comparing with other molecule on catalytic surfaces.

- it is clear that, in the sub-monolayer coverage of the DAP molecules, there exists hexagonal pores at the intersection of the wagon-wheel patterns (Fig. 3c). But it is unclear if at higher molecular coverages (Fig. 3b), these pores will be filled by molecules or not. Can the authors show some magnified STM images at higher coverages?

Reviewer #2 (Remarks to the Author):

The manuscript by He et al. claimed the self-assembly of 2,3-diaminophenazine (DAP) molecules porous network on MoSe₂, making use of the high density of mirror twin boundary (MTB) network in MBE-grown MoSe₂. The self-assembly is attributed to the slightly stronger interaction of the DAP molecules with the metallic MTB than with pristine MoSe₂ lattice, which also leads to two different types of aggregated configurations, namely the L-type and T-type. Overall, the formation of porous

network of the DAP molecules on MoSe₂ is interesting. However, I have a few technical comments the authors may want to further consider.

1. The authors claim that the L-type configuration preferentially forms on the MTB region while the T-type configuration dominates the pristine MoSe₂ lattice and the intersections of the MTB are left empty. A statistical analysis of the distribution of the two types of configurations on MTB regions and triangle domains would help to support the claim. In addition, can the authors comment on why the stronger interaction between DAP and MTB doesn't lead to exclusive (or at least preferential) adsorption of DAP on the MTB and why DAP doesn't adsorb at the intersections of the MTB?

2. The effect of self-assembly doesn't seem to be very prominent in this present study as compared to other literature reports on molecule self-assembly. Did the authors try different annealing conditions to improve the quality of the self-assembly? Specifically, it seems to me from Figure 3 and Figure S6 that the assembly of DAP in some regions of the sample doesn't look very ordered.

3. The discussion of angle-dependent adsorption of DAP on MTB doesn't seem to be supported by experimental results. A statistical analysis of the orientation distribution of DAP on MTB region would help to strengthen this part.

4. The idea of using the MTB network in MBE-grown MoSe₂ as a template for molecule self-assembly is interesting. Did the authors try different molecules and see if better assembly can be obtained?

Reviewer #3 (Remarks to the Author):

The manuscript by He, et al., reports on the formation of 2D porous network of organic molecules (DAP) following the net of mirror twin boundaries (MTBs) in single-layer MoSe₂ as a template. By taking advantage of the higher chemical reactivity of MTBs over the bare MoSe₂ surface, the molecular network naturally arranges in a hexagonal pattern. DFT calculations are carried out to provide insight regarding the formation mechanisms of the 2D molecular lattice.

Regarding the novelty and impact of the work, although some of the results presented will be of interest for the materials science community, I don't feel that none of them represent a significant advance in the field of 2D TMD materials and justifies its publication in Nature Communications. First, the extensive characterization of MTBs carried out by the authors is mostly reported in the literature. Second, the porous network is highly irregular and not continuous over entire the surface and, therefore, hardly scalable up to macroscopic sizes. Third, the network is self-assembled by weak non-covalent hydrogen bonds, which is detrimental of its mechanical stability as compared to existing 2D metal-coordination 2D covalent networks. This limits their potential to make these networks transferable. In my opinion, the main value of this work is the demonstration that this kind of 1D defects are susceptible to be chemically functionalized, which is nice result although not very surprising.

Point-by-point response to “Selective self-assembly of 2,3-diaminophenazine molecules on MoSe₂ mirror twin boundaries”

Reviewer #1 (Remarks to the Author):

Comment: *In this manuscript, Wee et al show the enhanced chemical reactivity of a special type of grain boundaries in transition metal chalcogenides (TMDs), namely the mirror-twin boundaries (MTBs) existing on MoSe₂.*

UHV STM clearly shows the atomic structure, electronic property and ‘local’ work function of MTBs in monolayer MoSe₂. The enhanced chemical reactivity of the MTBs is evidenced by the site-selective adsorption of 2,3-diaminophenazine (DAP) either parallel to or on top of the MTBs. Thus the molecular assembly could be steered and interesting patterns generated. The STM imaging and tunneling spectroscopy are of high quality, and the conclusions are adequately supported by complementary theoretical modeling; as such the study is pertinent and state of the art.

Overall, the ms reports several interesting findings, but key claims seem to be exaggerated. Notably “site-dependent electronic and chemical properties of MoSe₂” have been recognized earlier and “large-scale ordered nanostructures” remain elusive (see the ms abstract). Also on p.2 “regular porous networks” are mentioned, although no convincing data is at hand. It also must be said that line or other types of defects have been extensively employed to steer molecular assembly (using, e.g., vicinal surfaces, strain relief patterns, anisotropic crystal orientations, surface reconstructions, graphene or h-BN modulated templates etc.). Statements like ‘pioneering’ or ‘striking’ should be omitted and tangible ideas described towards improved control and how to use the realized nanosystems for applications.

Response: We thank the reviewer for the positive comments on our work being of high quality and interesting. We make the necessary revisions to address the concern that the “key claims seem to be exaggerated” below.

We agree that “the site-dependent electronic properties of MoSe₂ have been recognized earlier” both in experiment and theoretical calculations and we have cited the relevant references in our manuscript, e.g., ref. 4, 5, 11, 12, and 16. Although the atomic structures and metallic electronic properties of MoSe₂ mirror twin boundaries (MTB) have been reported by previous studies, the local work function (Fig. 2d) variations at different regions have not yet been reported. Moreover, the site-dependent chemical properties of MoSe₂, and also of other 2D TMDs, has not yet been studied. To the best of our knowledge, there are a few recent reports on selective adsorption of organic molecules on 2D TMDs studied by STM, e.g. dibenzothiophene (DBT) at the corner/edge sites of MoS₂ nanoclusters (ref. 18: *ACS Nano*, 2010, 4, 4677-4682), pentacene on 1H domains of 1H/1T patterned PtSe₂ (ref. 26: *Nat. Mater.* 2017, 16, 717-721), and TiOPc on the defected MoS₂ surface (ref. 23: *Sci. Adv.* 2017, 3, e1701661). Also, there is a study on preferential adsorption of Au clusters on the MoSe₂ MTBs (ref. 11: *ACS Nano* 2017, 11 5130). Other than these, molecular self-assembly steering by other types of defects have been reported in the literature, including “vicinal surfaces, strain relief patterns, anisotropic crystal orientations, surface reconstructions, graphene or h-BN modulated templates etc” as

suggested by the reviewer and discussed in our manuscript (page 2). However, there are no reports of systematic experimental investigations and theoretical studies of site-specific molecule self-assembly on 2D TMD line-defective surfaces.

The key finding of this study is the experimental realization of a DAP molecule porous network steered by the MoSe₂ MTB without the formation of covalent bonds. To understand the underlying mechanism, detailed AFM/STM analysis of the site-dependent properties of MoSe₂ and theoretical calculations were performed. The formation of the DAP porous networks extends over the entire MoSe₂ surface. The network sizes can reach hundred nanometers (Figure R1) and are only limited by the MoSe₂ island sizes. The limitations to the uniformity of the DAP porous nanostructures is due to disorder in the underlying MoSe₂ MTB network (as discussed in manuscript).

Figure R1. Large-scale STM images of 0.8 ML DAP molecule on MoSe₂. **a**, Size: 150 × 150 nm² (2.3 V, 90 pA); **b**, Size: 100 × 100 nm² (2.6 V, 60 pA). It shows that this DAP pores can extend over the MoSe₂ flakes.

The demonstration of site-dependent electronic and chemical properties of TMD monolayers indicates that such TMD surfaces could have applications in for site-selective catalysis. Catalytic molecules forming a patterned structure on MoSe₂ MTBs represent an organic/TMD hybrid network with selective catalytic functionality. Furthermore, this work would inspire further research on fabricating ordered defective nanostructures on 2D materials as templates for molecular self-assembly and other applications.

Revisions:

We have removed statements such as “large-scale ordered nanostructures”, “regular porous networks”, “pioneer” and “striking” in our manuscript as suggested by the reviewer. Figure R1 has also been added to the revised Supplementary Information (SI) (Figure S2). Some parts of the manuscript have also been revised accordingly.

Page 1, abstract: Our results demonstrate the site-dependent electronic and chemical properties of MoSe₂ monolayers, which can be exploited as to be a natural template to create large-scale ordered nanostructures.

Page 2, 1st paragraph: We observed the formation of a regular porous network of DAP

molecules that map onto the wagon-wheel patterns of the underlying MoSe₂.

This **pioneering** study demonstrates that organic molecule self-assembly can be facilitated by domain boundaries in epitaxial 2D TMDs.

The whole MoSe₂ surface is decorated by **striking** triangle patterns.

Page 4, 3rd paragraph: The adsorbed DAP area expand to fill the whole ML-MoSe₂ surface with increasing coverage as shown in Fig. 3b. **The network sizes can reach hundred nanometers (Figure S2) and are only limited by the MoSe₂ island sizes.**

Comment: *Furthermore, there are some technical issues, see comments below:*
- *page 3 Paragraph 2: “bright contrast (higher Δf) is generally a result of short-range Pauli repulsive interactions, while darker contrast (lower Δf) is due to long-range”; this description is not accurate.*

The AFM repulsive regime should correspond to positive Δf , and the attractive regime should correspond to negative Δf , just calling it higher or lower without mentioning the sign of Δf is misleading.

Response: Thanks for correcting our description. The following sentences have been revised.

Revisions:

Page 3, 2nd paragraph: “In frequency-shift nc-AFM images which are sensitive to the tip-sample separation as well as the local electron densities, **brighter contrast (with higher Δf) positive Δf** is generally a result of short-range Pauli repulsive interactions, while **darker contrast (with lower Δf) negative Δf** is due to long-range attractive van der Waals interactions and/or electrostatic forces³¹. Therefore, the observed **lower frequency darker contrast with negative frequency** shift in the MTB regions might be due to their relatively lower height and/or rich electron densities in comparison to the MoSe₂ triangle domains.”

Comment: - *on page 6 Paragraph 1, the authors mentioned “the adsorption energy of the DAP molecule adsorbed on the MTB is 0.16 eV larger than that on the defect-free MoSe₂”. Can the authors comment on how useful this energy difference is, (for example) by comparing with other molecule on catalytic surfaces.*

Response: The energy difference of 0.16 eV between the DAP adsorbed on the MTB and the defect-free region is not large. However, this small energy difference is enough to drive the site-selective adsorption of the DAP molecules on the surface as demonstrated in the manuscript. DAP molecules are physisorbed on both MTBs and MoSe₂ domains with binding energies of -1.38 eV and -1.22 eV respectively, which are relatively weak compared to other molecules on metallic catalytic surfaces such as Pt(111) and Cu(111). Even for organic molecule adsorbed on metallic TMD materials, the binding energy can be over 10 eV. For example, theoretical calculations suggest that pentacene molecules adsorb on 2H- and 1T- MoS₂ monolayer with binding energies of -1.6 eV and -13.9 eV respectively (*Adv. Mater. Interfaces* 2017, 4, 1601083). The 2H MoS₂ is semiconducting while the 1T phase is metallic. The -1.6 eV binding energy for pentacene on semiconducting 2H-MoS₂ is comparable to that for DAP on MoSe₂ in our study. Note that the adsorption energy of a molecule on a surface and the energy to trigger a catalytic

reaction are different. Many TMD materials, particularly MoS₂, are good catalysts. For example, MoS₂ has been used as a catalyst for desulfurization in petrochemistry and hydrogenation for organic synthesis. Some catalytic reactions are suggested to occur at the S-vacancy and edges. As reported by Yu et al (*Nat. Commun.*, 2014, 5, 5290), (3-mercaptopropyl) trimethoxysilane (MPS) molecules can preferentially adsorb to S vacancies on the MoS₂ surface, and the S-C bond of the thiolate molecule could dissociate and leave the lone S atom in the vacancy site after thermal annealing. The energy barrier for the S-C bond dissociation is only 0.22 eV.

On page 2, we suggest that “These porous organic molecule networks on TMDs have potential applications in site-selective catalysis, or as molecular sensors or flexible organic optoelectronic devices.” The selective self-assembly on MoSe₂ MTBs demonstrates that defective sites on TMDs could be used for site-selective catalysis.

Comment: *it is clear that, in the sub-monolayer coverage of the DAP molecules, there exists hexagonal pores at the intersection of the wagon-wheel patterns (Fig. 3c). But it is unclear if at higher molecular coverages (Fig. 3b), these pores will be filled by molecules or not. Can the authors show some magnified STM images at higher coverages?*

Response: With increasing molecular coverage, close-packing phases will emerge. As shown in Figure R2, we find that the coexistence of the hexagonal porous network (region B) with the densely close-packing phase (region A). From the inset line profile in Figure R2b, we can see that the height for the molecules is ~ 6 Å in region A, while ~2 Å only in region B. This reveals that the orientation of DAP molecules transits from flat-laying in the porous network (region B) to standing-up (with a tilt angle of ~60° between the molecular long axis and the substrate surface) in the closed packing crystalline phase (region A). The molecular model of the closed packing phase (region A) is shown in Figure R2d (top view) and 2e (side view), which is close to the packing structure for DAP bulk crystal (ref. 28: *Acta Cryst.* 2001, C57 104-105). This agrees with our calculation results (ref. Page 5-6, 2nd and 3rd paragraph) that the flat-lying DAP molecules are weakly physisorbed on the MoSe₂ surface, and the close-packing configuration and the bulk crystalline structure are stabilized by intermolecular π - π interactions and hydrogen bonding.

Figure R2. **a**, Large-scale STM image (2.3 V, 90 pA; $200 \times 200 \text{ nm}^2$) of 1.2 ML DAP on MoSe₂. Region A, B, C and D highlight the close packing monolayer DAP on monolayer MoSe₂, sub-monolayer DAP porous network on monolayer MoSe₂, monolayer DAP on HOPG and sub-monolayer DAP on the bilayer MoSe₂ islands, respectively. **b**, Magnified STM image (2.3 V, 70 pA; $25 \times 25 \text{ nm}^2$) of the region A and B. The insert is the line-profile corresponding to the blue line AA'. It shows that the height of sub-monolayer DAP porous network is about 207 pm, while the close-packing DAP monolayer is about 805 pm. **c**, High resolution STM (2.0 V, 70 pA; $10 \times 10 \text{ nm}^2$) image of region A. The unit cell is highlighted by a white rectangle, with $a = 0.944 \pm 0.05 \text{ nm}$ and $b = 1.129 \pm 0.035 \text{ nm}$. **d-e**, Schematic models of the inclined packing structure of monolayer DAP on MoSe₂ in region A.

Revisions:

Following the reviewer's suggestion, we have included Figure R2 as supplementary Figure S6 in the revised supplementary information, and also the above discussion. Some parts of the manuscript have also been revised accordingly.

Page 5: "With increasing molecular coverage, a close-packed phase will emerge coexisting with the loose-packed porous phase (Figure S6)."

Reviewer #2 (Remarks to the Author):

Comment: *The manuscript by He et al. claimed the self-assembly of 2,3-diaminophenazine (DAP) molecules porous network on MoSe₂, making use of the high density of mirror twin boundary (MTB) network in MBE-grown MoSe₂. The self-assembly is attributed to the slightly stronger interaction of the DAP molecules with the metallic MTB than with pristine MoSe₂ lattice, which also leads to two different types of aggregated configurations, namely the L-type and T-type. Overall, the formation of porous network of the DAP molecules on MoSe₂ is interesting. However, I have a few technical comments the authors may want to further consider.*

The authors claim that the L-type configuration preferentially forms on the MTB region while the T-type configuration dominates the pristine MoSe₂ lattice and the intersections of the MTB are left empty. A statistical analysis of the distribution of the two types of configurations on MTB regions and triangle domains would help to support the claim. In addition, can the authors comment on why the stronger interaction between DAP and MTB doesn't lead to exclusive (or at least preferential) adsorption of DAP on the MTB and why DAP doesn't adsorb at the intersections of the MTB?

Response: We thank the reviewer for the constructive suggestions. We have analyzed ~1000 DAP molecules to understand the distributions of the L-type and T-type configurations and the statistical histogram is shown in Figure R3. The percentage of L-type configurations, 58.5%, is higher than that of the T-type. This observation is in contrast to the relatively higher formation energy per molecule for the T-type configuration (-0.21 eV) than the L-type (-0.12 eV) without considering their adsorption sites. As the L-type configurations are preferentially adsorbed atop the MTBs and the T-types on the MoSe₂ domains, the slightly preferential L-type configuration is attributed to the stronger interactions between the DAP molecules and the MTBs.

Figure R3. A statistical analysis of the distribution of the two types of configuration of DAPs (~1000 DAP molecules). The percentage of L-type configuration is higher than that of T-type.

To understand why molecules do not adsorb at the intersections of the MTBs, we had carried out STM/STS and AFM measurements to investigate the atomic and electronic structures of the intersections. As shown in Figure R4a-d, various types of intersections can be found in our MoSe₂ samples. This is consistent with the observations of at least six different types of MTB intersections in a previous ACS nano paper [ref. 11: *ACS nano* 11, 5130 (2017)]. From the atomic models, we note that the intersections are usually Se-rich. The intersections also have a larger bandgap than the defect free MoSe₂ region. These reasons suggest why DAP does not adsorb at the intersections of the MTBs. However, to fully answer this question, more studies are required, e.g., DFT calculations, which are beyond the scope of the present study.

Figure R4. a-d, High-resolution nc-AFM images of different MTB intersections (size: $5 \times 5 \text{ nm}^2$) and the corresponding atomistic models. e, dI/dV spectra recorded at the defect-free MoSe₂ domain (red line) and the intersection (green line) respectively.

Revisions:

Figure R3 has included as supplementary Figure S5 in revised supplementary information, as well as the above discussion. Some parts of the manuscript have also been revised accordingly.

Page 5: In the porous network, two typical packing configurations are observable, namely linear configurations (L-type) and triangle trimer configurations in a head-to-tail arrangement (T-type) highlighted by the green circles and red triangles in Fig. 3a respectively. We have analyzed ~1000 DAP molecules to understand the distributions of the L-type and T-type configurations (the statistical histogram is shown in Figure S5). The percentage of L-type configurations is slightly higher than that of the T-type.

Page 7: In the absence of a substrate, DFT calculations suggest that DAP molecules in the T-type configuration ($E_{\text{bind}} = -0.21$ eV) is more stable than that in the L-type configuration ($E_{\text{bind}} = -0.12$ eV) due to the stronger hydrogen bonding interactions, which is in contrast to the slightly higher distribution of the L-type configurations as analysed in Figure S5.

Comment: *The effect of self-assembly doesn't seem to be very prominent in this present study as compared to other literature reports on molecule self-assembly. Did the authors try different annealing conditions to improve the quality of the self-assembly? Specifically, it seems to me from Figure 3 and Figure S6 that the assembly of DAP in some regions of the sample doesn't look very ordered.*

Response: The porous DAP molecular network is distinctly formed by site-specific self-assembly: preferential T-type adsorption on the defect-free MoSe₂ domains and L-type on the MTBs, while leaving the MTB intersections empty. This porous nanostructure is not observable for the DAP molecules grown on graphite and Au(111) substrate. On graphite, the DAP molecules assemble into close-packed ordered arrays as shown in Figure R5; while on Au(111), the DAP molecules assemble into chain-like structures at low coverage and a disordered packing structure at higher coverage, as shown in Figure R6.

Figure R5. DAP molecules on HOPG. **a**, Large area STM image (2.7 V, 60 pA; 100×100 nm²) of monolayer DAP on HOPG. α , β and γ phase show the dense close-packing, spare close-packing and disorder structure. **b**, The line-profile corresponding to the blue line AA' in panel a. The height of monolayer DAP molecules on HOGP is about 310 pm. The height of α and β phase almost are same, indicating that both of them are flat lying on the HOPG substrate. **c-d**, Magnified STM image (2.7 V, 60 pA; 14 × 7 nm²) and schematic diagram of α phase. Here, $a = 1.24 \pm 0.05$ nm, $b = 1.17 \pm 0.05$ nm. **e-f**, Magnified STM image (2.3 V, 60 pA; 14 × 7 nm²) and schematic diagram of β phase. Here, $a = 1.249 \pm 0.05$ nm, $b = 1.076 \pm 0.05$ nm.

Figure R6. DAP molecules on Au(111). **a**, STM image (-0.98 V, 60 pA; $20 \times 20 \text{ nm}^2$) of DAP molecules on Au at low coverage. It shows a chain-like structure due to intermolecular hydrogen bonding. **b**, STM image (-1.06 V, 60 pA; $20 \times 20 \text{ nm}^2$) of monolayer DAP on Au(111) shows irregular packing structure.

The relatively weak selective adsorption in this study is attributed to: (1) the MoSe₂ MTB networks are not uniform (the distribution of the MoSe₂ domain sizes is shown in Fig. 1b); (2) the relatively weak interactions between the DAP molecules and the MoSe₂ (DAP molecules are physisorbed both on the defect-free domain and MTB region). The first contribution could be the dominant reason for the poorer ordering and uniformity of the porous DAP networks. By comparing Fig. 1b and Fig. 1d, we can see that the formation of the hexagonal DAP porous structures is steered by the underlying MTBs.

We have studied the evolution of the DAP packing structure with annealing temperature. We found that the DAP porous nanostructures do not become more ordered after thermal annealing, desorb above 90°C. As shown in Figure R7, the DAP pores did not change when annealing temperature was lower than 60°C, and the sub-monolayer DAP molecules will desorb from the MoSe₂ surface at 90°C, while the first-layer DAP molecules with close-packing configuration only start to desorb at 140°C.

Figure R7. **a**, STM image (2.3 V, 70 pA; $120 \times 120 \text{ nm}^2$) of 1.1 ML DAP molecule on the MoSe₂ surface before annealing. **b-d**, The DAP samples after annealing at 60°C, 90°C, and 140°C respectively. Region A, B and C mark the close-packing DAP on MoSe₂, porous DAP monolayer on MoSe₂ and monolayer DAP on HOPG. In panel **b**, both the porous DAP networks and the close-packing structures are well resolved after annealing at 60°C. In panel **c**, the porous networks disappear while the close-packing structures remain upon annealing at 90°C. In panel **d**, the close-packed DAP monolayer also

desorb leaving the MoSe₂ empty after annealing at 140°C. (**b**, 2.5 V, 60 pA, 100 × 100 nm²; **c**, 2.5 V, 60 pA; 100 × 100 nm²; **d**, -2.6 V, 80 pA; 80 × 80 nm²).

Revisions:

We have included Figure R5–R7 as supplementary Figure S8, S9, S7, and above discussion is added to the revised manuscript and SI. Some parts of the manuscript have also been revised accordingly.

Page 5-6: The porous DAP molecular network is distinctly formed by a site-specific self-assembly: preferential T-type adsorption on the defect-free MoSe₂ domains and L-type on the MTBs, while leaving the MTB intersections empty. With increasing molecular coverage, a close-packing phases will emerge coexisting with the loose-pacing porous phase (Figure S6). The evolution of the DAP packing structure with annealing temperature have also been studied. We find that the DAP porous nanostructures do not become more ordered after thermal annealing, but desorb above 90°C (Figure S7). The close-packing configuration start to desorb at 140°C. This porous nanostructure is not observable for the DAP molecules grown on graphite and Au(111) substrate at various coverages (Figure S8-S9). Therefore, the formation of the hexagonal DAP porous structures is steered by the underlying MTBs, and the relatively weak selective adsorption attributed to the subtle competition among the relatively weak DAP-substrate and DAP-DAP interactions.

Comment: *The discussion of angle-dependent adsorption of DAP on MTB doesn't seem to be supported by experimental results. A statistical analysis of the orientation distribution of DAP on MTB region would help to strengthen this part.*

Response: We thank the reviewer for this constructive suggestion. We have summarized the angle-dependent adsorption of DAP on MTBs in Figure R8, and included it as supplementary Figure S4 in revised supplementary information, which could help to strengthen this part.

Figure R8a is the schematic of the definition of the rotation angle (θ) of DAP relative to MTBs, where the grey arrows highlight the MTB orientations, and the blue arrow lines mark the long axis of DAP molecules. The MTBs are three-fold symmetric and have three equivalent directions, and the DAP adsorption positions (i.e., the molecular centre) are usually centred away from the MTBs (see Figure 4b and d). For this statistical analysis, we only consider the orientation of the selected molecule relative to one selected MTB orientation. To be consistent with the theoretical model, the statistical step is set to 10° (e.g., 0 - 10°, 10 - 20°, etc.). From the histogram, we can see that the molecules orientated at ~0° are dominant (~30.8%), compared to the other orientations. This result agrees with the calculated angle-dependence adsorption energy of DAP molecule adsorbed on MTB shown in Figure 6, which suggests that the molecule adsorbed parallel (0°) to the MTB with lowest binding energy. The relatively weak preference agrees with the fact that the calculated binding energy difference to the other orientations are not large, i.e., < 0.16 eV.

Figure R8. a, The schematic of the rotation angle (θ) of DAP relative to MTBs. The grey arrows show the directions of MTBs; the blue arrow shows the long axis of DAP molecules. **b**, A statistical analysis of the orientation distribution of DAP (more than 800 DPA molecules) relative to MTBs. From the histogram, we can see that the molecules orientated at $\sim 0^\circ$ are dominant ($\sim 30.8\%$), compared to other orientations.

Revisions:

Page 5: By a statistical analysis of the orientation distribution of the DAP molecules with respect to the MTBs (Figure S4), it is found that the DAP molecules oriented with their long axes parallel to the MTBs are dominant. The alignment of the molecular orientations with the three high symmetry directions of MoSe₂ indicating a significant substrate effect⁴³.

Page 7: The relatively small energy difference (< 0.16 eV) between the most stable configuration and the others is consistent with the molecular orientation distribution observed in Fig. 4 and Fig. S4.

Comment: *The idea of using the MTB network in MBE-grown MoSe₂ as a template for molecule self-assembly is interesting. Did the authors try different molecules and see if better assembly can be obtained?*

Response: We thank the reviewer for this constructive suggestion. We had deposited different molecules onto the MoSe₂ MTBs, including HATCN (Figure R9a-b), GaClPc (Figure R9c-d) and DAP. As shown in Figure R9a, the HATCN molecules preferentially adsorb on HOPG surface to form bilayer films (R1) due to strong π - π interactions, leaving the MoSe₂ surface empty. For GaClPc molecules, single-layer and bilayer close-packing islands can be found on both the HOPG and MoSe₂ surface, similar to the growth behavior of other dipole Pc molecules, e.g., ClAlPc on HOPG (*Phys. Rev. B*, 87, 085205, 2013). Only the DAP molecules demonstrate distinctly selective adsorption behaviour with the MTBs. Therefore, the selection of the molecule is very important in order to investigate the site-dependent properties of the MTB networks. The relatively chemical reactivity of the DAP amine groups could play an important role in the site-selectivity. As revealed by theoretical calculations in Figure 5, the charge redistribution is more pronounced at the DAP amino groups. Another important reason may be due to the moderate molecule-molecule and molecule-substrate interactions for the DAP system. For HATCN, the π - π molecule-graphite and molecule-molecule interactions are dominant; while for GaClPc molecule, the dipole-dipole and also π - π interactions overwhelm the difference between different adsorption sites.

Figure R9. a-b, 1.2 ML HATCN molecule on MoSe₂/HOPG surface. **a**, Large-scale STM image (2.9 V, 50 pA; 100 × 100 nm²) show that the HATCN molecules only adsorb on HOPG surface to form bilayer films (R1) due strong π - π interactions. **b**, Magnified STM image (-1.5 V, 105 pA; 7 × 7 nm²) of the empty MoSe₂ surface (R2) corresponding to the red rectangular region in panel **a**. The inset in panel **a** is the schematic structure of HATCN molecule. **c-d**, 1.5 ML GaClPc molecule on MoSe₂/HOPG surface. **c**, Large STM image (1.8V, 100 pA; 150 × 150 nm²) shows that both MoSe₂ and HOPG surface are fully covered with GaClPc molecules. R3 and R4 show single-layer and bilayer GaClPc regions on HOPG respectively. R5 and R6 show the disordered single-layer and well-ordered bi-layer GaClPc molecules on MoSe₂ respectively. The inset in panel c is the structure of GaClPc molecule. **d**, Magnified STM image (-1.2V, 100 pA; 10 × 10 nm²) corresponds to the red rectangular region in panel **c**, showing the close-packed GaClPc bilayer on MoSe₂ surface.

Reviewer #3 (Remarks to the Author):

Comment: *The manuscript by He, et al., reports on the formation of 2D porous network of organic molecules (DAP) following the net of mirror twin boundaries (MTBs) in single-layer MoSe₂ as a template. By taking advantage of the higher chemical reactivity of MTBs over the bare MoSe₂ surface, the molecular network naturally arranges in a hexagonal pattern. DFT calculations are carried out to provide insight regarding the formation mechanisms of the 2D molecular lattice.*

Regarding the novelty and impact of the work, although some of the results presented will be of interest for the materials science community, I don't feel that none of them represent a significant advance in the field of 2D TMD materials and justifies its publication in Nature Communications. First, the extensive characterization of MTBs carried out by the authors is mostly reported in the literature. Second, the porous network is highly irregular and not continuous over entire the surface and, therefore, hardly scalable up to macroscopic sizes. Third, the network is self-assembled by weak non-covalent hydrogen bonds, which is detrimental of its mechanical stability as compared to existing 2D metal-coordination 2D covalent networks. This limits their potential to make these networks transferable. In my opinion, the main value of this work is the demonstration that this kind of 1D defects are susceptible to be chemically functionalized, which is nice result although not very surprising.

Response: We thank the reviewer for highlighting the novelty of the selective chemical functionalisation of the 1D defects. We think this work is important for the following reasons.

Firstly, the key finding of this study is the experimental realization of a DAP molecule porous network steered by the higher chemical reactivity of the MoSe₂ MTBs. In order to understand the underlying mechanism, detailed analysis of the atomic and electronic properties of the MoSe₂ MTB network has been performed. (1) The statistical analysis of the triangle MoSe₂ domain sizes shown in Figure 1b provides strong evidence of the template effect of the MoSe₂ MTB network for the DAP self-assembly. (2) Our nc-AFM/STM measurements were performed at 77 K by a tungsten tip without CO-functionalization. The high resolution AFM images shown in Figure 1d (30 × 30 nm²) and Figure 1e (5 × 5 nm²) indicate that the nc-AFM technique can be applied to characterize 2D TMD materials without liquid helium and CO. (3) Although the atomic structures and metallic electronic properties of MoSe₂ MTBs have been reported by previous AFM/STM studies, the local work functions (Fig. 2d) at different regions is a new result.

Secondly, the formation of the DAP porous networks can extend over the MoSe₂ islands, and the network sizes can reach over hundred nanometers (Figure R1 or S2). The sizes of DAP islands are only limited by the MoSe₂ island sizes. The lack of uniformity of the porous nanostructures is mainly due to the lack of uniformity of the underlying MoSe₂ MTB networks (as discussed in the manuscript). We believe the publication of our work would inspire further interesting research on highly ordered defective nanostructures on 2D materials at larger scales, for use as templates for molecular self-assembly and other applications.

Thirdly, the formation of the observed DAP porous networks is steered by site-specific molecule-substrate interactions and non-covalent intermolecular hydrogen bonds. We acknowledge the mechanical stability of this network is not comparable to other 2D metal-coordination covalent networks. However, this study aims to investigate the site-dependent electronic and chemical properties of MoSe₂ MTB networks, and the selected DAP molecules demonstrate stronger interactions at the MTB defects rather than the defect-free regions. For future applications, we hope that such porous organic networks on TMDs could have potential uses as molecular sensors or in flexible organic optoelectronic devices. If the semiconducting TMDs are prepared on insulator substrates (e.g., SiO₂/Si), there will be no need to transfer the organic networks. To realize these aims, more research is required, including the fabrication of

large-scale high quality TMDs with uniform defects and the design of organic molecules with desired functionalities.

With the above clarifications and the corresponding modifications in the revised manuscript and SI, we hope that the novelty and the impact of this work is better communicated.

Revisions:

Page 3: The nc-AFM measurements were carried at 77 K with a native tungsten tip without CO functionalization. Fig.1d shows the nc-AFM image of the same area as the STM image in Fig. 1c with high resolution.

Page 4-5: The adsorbed DAP area expand to fill the whole ML-MoSe₂ surface with increasing coverage as shown in Fig. 3b. The network sizes can reach hundred nanometers (Figure S2) and are only limited by the MoSe₂ island sizes.

Page 5-6: The porous DAP molecular network is distinctly formed by a site-specific self-assembly: preferential T-type adsorption on the defect-free MoSe₂ domains and L-type on the MTBs, while leaving the MTB intersections empty. With increasing molecular coverage, a close-packed phase will emerge coexisting with the loose-packed porous phase (Figure S6). The evolution of the DAP packing structure with annealing temperature has also been studied. We find that the DAP porous nanostructures do not become more ordered after thermal annealing, but desorb above 90°C (Figure S7). The close-packed configuration starts to desorb at 140°C. This porous nanostructure is not observable for the DAP molecules grown on graphite and Au(111) substrate at various coverages (Figure S8-S9). Therefore, the formation of the hexagonal DAP porous structures is steered by the underlying MTBs and the relatively weak selective adsorption attributed to the subtle competition among the relatively weak DAP-substrate and DAP-DAP interactions.

REVIEWERS' COMMENTS:

Reviewer #1 supplied comments to the editor.

Reviewer #2 (Remarks to the Author):

The authors have addressed most of my technical comments. While the results are overall interesting, it remains, as also acknowledged by the authors and pointed out by the other two referees, that the effect of self-assembly may be a bit exaggerated as the porous network is not as ordered as one would expect from self-assembly and is quite small in size. On the positive side, this study may stimulate further studies on molecular self-assembly using defect networks in 2D materials as templates.

Point-by-point response to “Selective self-assembly of 2,3-diaminophenazine molecules on MoSe₂ mirror twin boundaries”

Reviewer #2 (Remarks to the Author):

Comment: *The authors have addressed most of my technical comments. While the results are overall interesting, it remains, as also acknowledged by the authors and pointed out by the other two referees, that the effect of self-assembly may be a bit exaggerated as the porous network is not as ordered as one would expect from self-assembly and is quite small in size. On the positive side, this study may stimulate further studies on molecular self-assembly using defect networks in 2D materials as templates.*

Response: We thank the suggestions from the editor and the reviewers. As they all shared the same sentiment, we consolidate the responses together. According to the reviewer’s suggestion, we have used the “porous structure” instead of “porous network” in the revised version. All the revised sentences are listed below:

Page 1, Abstract: In this work, we demonstrate the self-assembly of 2,3-diaminophenazine (DAP) molecule porous ~~network~~–~~structure~~ with alternate L-type and T-type aggregated configurations on the MoSe₂ hexagonal wagon-wheel pattern surface.

Page 2, 1st paragraph: We observed the formation of a porous ~~network~~–~~structure~~ of DAP molecules that map onto the wagon-wheel patterns of the underlying MoSe₂.

Page 2, 1st paragraph: These defective TMDs and porous organic molecule ~~networks~~–~~structures~~ have potential applications in site-selective catalysis, or as molecular sensors or flexible organic optoelectronic devices.

Page 4, 3rd paragraph: Upon deposition of a sub-monolayer of DAP molecules, the supramolecular porous ~~networks~~–~~structures~~ formed are obvious and distinguishable, but their sizes and shapes are not uniform as shown in Fig. 3a-b.

Page 4, 3rd paragraph: The ~~network~~–~~porous structure~~ sizes can reach hundred nanometers (Figure S2) and are only limited by the MoSe₂ island sizes.

Page 5, 1st paragraph: Fig. 4a shows a high resolution STM image of the DAP ~~nanoporous network~~–~~structure~~ with a partially uncovered MoSe₂ surface.

Page 5, 2nd paragraph: In the porous ~~network~~–~~structure~~, two typical packing configurations are observable, namely linear configurations (L-type) and triangle trimer configurations in a head-to-tail arrangement (T-type) highlighted by the green circles and red triangles in Fig. 3a respectively.

Page 5, 2nd paragraph: The hexagonal DAP ~~networks~~–~~porous~~ structures form surrounding the intersection of wagon-wheel patterns by alternatingly absorbing the L-type and T-type configurations.

Page 5, 3rd paragraph: The porous DAP molecular ~~network~~–~~structure~~ is distinctly formed by a site-specific self-assembly:

Page 6, 1st paragraph: To gain further insight into the formation mechanism of the DAP porous ~~network~~–~~structure~~, theoretical calculations based on density functional theory (DFT) were performed to

investigate the templating effect of the MTB network.

Page 7, 2nd paragraph: Finally, the formation of the DAP porous network-structure, particularly the coexistence of T-type and L-type building configurations in Fig. 4, can be understood from a comparison of their formation energies (E_{bind}).

Page 8, 1st paragraph: In conclusion, we demonstrate the self-assembly of a DAP molecular nanoporous network-structure on the dense MTB network surface of an MBE-grown MoSe₂ film.

Page 8, 1st paragraph: Thus, the formation of the DAP porous network-structure is guided by the MoSe₂ MTB network together with intermolecular hydrogen interactions.

Page 11: Fig. 3 STM images of the sub-monolayer DAP on MoSe₂/HOPG. **a** 0.36 ML and **b** 0.80 ML DAP molecules assemble on the MoSe₂ surface, forming porous supramolecular network-structure.

Page 11: Fig. 4 Model of DAP porous configurations on MoSe₂ surface. **a** High resolution STM image ($15 \times 15 \text{ nm}^2$; $V_s = 1.7 \text{ V}$, $I = 60 \text{ pA}$) of DAP porous network-structure on MoSe₂.